# The Associations between Obsessive Compulsive Personality Traits, Self-Efficacy, and Exercise Addiction

**DOI:** 10.3390/bs13100857

**Published:** 2023-10-19

**Authors:** Catherine So Kum Tang, Kai Qi Gan, Wai Kin Lui

**Affiliations:** 1Department of Counselling & Psychology, Hong Kong Shue Yan University, 10 Wai Tsui Crescent, Braemar Hill, North Point, Hong Kong, China; 2Department of Psychology, National University of Singapore, 21 Lower Kent Ridge Road, Singapore 119077, Singapore

**Keywords:** exercise addiction, obsessive compulsive personality traits, self-efficacy, behavioral addiction, young adults

## Abstract

Exercise addiction refers to maladaptive exercise patterns involving compulsivity and addiction-like behaviors. Exercise addiction has been found to relate to negative physical and mental health outcomes such as heart abnormalities, physical injuries, and interpersonal conflicts. Based on the social cognitive theory, this study investigated the extent to which the interplay of obsessive compulsive personality disorder (OCPD) traits and self-efficacy beliefs would influence the development and maintenance of exercise addiction. A total of 1228 college students in the United States responded to an online survey. Based on cut-off scores of the Revised Exercise Addiction Inventory, the prevalence estimates of exercise addiction were 4.0% for males and 1.8% for females. Results showed that males are more prone to exercise addiction than females. Also, OCPD traits and self-efficacy significantly predicted exercise addiction after controlling for age and sex. Self-efficacy acted as a moderator in influencing the relationship between OCPD traits and exercise addiction, especially for females. At high levels of self-efficacy, more OCPD traits were significantly associated with a higher risk of exercise addiction. However, at low levels of self-efficacy, there was no association between OCPD traits and exercise addiction. The findings suggest that public education and intervention for exercise addiction should attend to the interplay between personality factors and sex.

## 1. Introduction

Exercise addiction refers to excessive exercise characterized by six common components of addiction, namely salience, mood modification, tolerance, withdrawal, personal conflict, and relapse [1,2,3]. While regular exercise is generally a healthy activity, exercise addiction involves performing excessive amounts of exercise to the detriment of physical health, spending excessive time exercising to the detriment of one’s personal and professional life, and exercising despite physical injuries [4,5,6]. In addition, exercise addiction involves a state of dependence upon regular exercise with the occurrence of severe withdrawal symptoms when unable to exercise [7,8]. Individuals with a higher risk of exercise addiction also tend to be more depressed and anxious when compared with individuals with a lower risk of exercise addiction [8,9,10]. Moreover, exercise addiction frequently co-occurs with eating disorders, body image disorders, and obsessive compulsive disorder [4,11,12].

Exercise addiction has not been listed as a clinical disorder in the Diagnostic and Statistical Manual of Mental Disorders yet [13]. Current prevalence estimates of exercise addiction vary significantly, likely due to the use of different assessment measures and differences in samples used [11,12,14,15]. Most studies have focused on identifying the prevalence of exercisers at risk of exercise addiction, using the Exercise Addiction Inventory (EAI) [2], the Exercise Addiction Scale (EDS) [16], or the Obligatory Exercise Questionnaire (OEQ) [17]. These studies have found prevalence rates ranging from 3% to 9% among regular exercisers [18,19,20], 7.1% among football players, and 9.7% among fitness club exercisers [21]. Among the general adult population, an estimated prevalence rate of 0.3–0.5% in a nationally representative study conducted in Hungary was found [22]. Significantly higher prevalence rates were found in populations of undergraduate college students in Australia (41.3% for males; 26.5% for females) [23], likely due to the use of the OEQ. Two meta-analyses found that, as compared with the EAI and EDS, the OEQ returned higher prevalence estimates as it measures different aspects of exercise addiction [12,15]. In general, being male and younger tend to be associated with a greater risk of exercise addiction [19,24,25].

### 1.1. Theoretical Framework: Social Cognitive Theory

The social cognitive theory (SCT) [26,27] has been used as a framework to study human physical activity, sport performance, exercise adherence, etc. [28]. SCT uses the concept of reciprocal determinism, where behavioral, individual, and environmental factors interact with each other to produce behavior changes. Whether a particular behavior will be adopted also depends on the anticipated consequences of that behavior. Guided by SCT, the present study examined the extent to which the interplay of individual factors of personality traits, specifically obsessive compulsive personality disorder traits, and self-efficacy beliefs influence the development and maintenance of exercise addiction.

### 1.2. Obsessive Compulsive Personality Disorder (OCPD) Traits

Personality traits such as perfectionism and narcissism have been found to associate with exercise addiction, and these associations are especially robust in the presence of obsessive compulsiveness [29]. Obsessive compulsive personality disorder (OCPD) is characterized by excessive concern with orderliness, perfectionism, attention to details, mental and interpersonal control, and a need to control one’s environment [13]. Research on clinical populations with eating disorders generally shows a positive relationship between OCPD traits and exercise addiction [30,31]. There are several plausible explanations regarding the relationship between OCPD traits and exercise addiction. OCPD is closely linked to perfectionism [32,33,34], which is in turn related to a higher risk of exercise addiction [5,35,36]. One of the biggest motivations to engage in exercise is a need to adhere to societal expectations for one’s appearance and perceived physical healthiness, resulting in a strong drive for thinness and/or muscle gain for both males and females [37,38,39,40,41,42,43]. Furthermore, the use of social media may reinforce distorted thoughts regarding an ideal body image and maladaptive behaviors such as excessive exercise [37,40,43]. Individuals with more OCPD traits may feel the need to achieve the ‘ideal’ figure at any cost, and thus set unrealistically high expectations for themselves [44].

Rigidity, an important feature of OCPD [12,45,46], is reflected in the exercise schedules of individuals who are at a high risk of exercise addiction [44,47,48]. Even when injuries occur, these individuals refuse to make changes to their exercise regimes. This rigidity surrounding their exercise regimes tends to contribute to the development of exercise addiction [48]. This rigidity is also observed in relatively low levels of exercising, such as feeling the need to do 30 min’ worth of daily weight training even when fatigued [44,47]. Only when they complete their regimes do they feel better, less frustrated, and less anxious. In addition, individuals with OCPD feel a need to be in control [45], and exercise helps to provide a sense of perceived self-control [5,49]. Hence, individuals with OCPD traits are prone to exercise addiction.

### 1.3. Self-Efficacyche

Self-efficacy is one of the main tenets of SCT and is defined as individuals’ belief in their own capabilities to perform certain actions, operating within the interactional structure of individual, environmental, and behavioral factors [26,27]. According to SCT, self-efficacy has a direct influence on behavior, as well as an indirect influence via goal setting [50]. A review study found that research across life settings provided support for the latter contention; that is, people who are more efficacious tend to set more challenging goals and persist to a greater extent in the face of adversity [28]. When individuals set their own goals for physical activity, they tend to increase their physical activity behaviors [51]. Self-efficacy has also been implicated as a reliable predictor of both exercise adoption and adherence [28,52,53,54].

Exercise addiction involves excessive amounts of exercise, which is an effortful activity. Thus, to be able to persist in large amounts of exercise despite its effects, such as exhaustion, requires a certain level of self-efficacy. Individuals who are addicted to exercise have to believe in their own ability to continue their efforts to exercise despite its effects or else they would have quit already. Hence, these individuals should have a relatively high level of self-efficacy to keep up with their exercise regimes and to be addicted to exercise. In other words, high levels of self-efficacy should be related to a risk of exercise addiction. Indeed, it was found that the better an individual perceives themselves to be at his/her sport, or in other words, the greater the self-efficacy one has regarding his/her sport, the greater the risk of exercise addiction one may have [35]. An alternative perspective is that self-efficacy may interact with specific personality traits to bring about maladaptive behaviors. Considering that it is somewhat contradictory for higher levels of self-efficacy to be associated with maladaptive behaviors, researchers suggested that whether high self-efficacy leads to adaptive or non-adaptive behaviors depends on the presence of other factors [35]. For instance, if individuals are highly self-critical, they will less likely be easily satisfied by their accomplishments. High levels of self-efficacy ensure that these individuals do not give up even when they are affected physically and mentally for the sake of greater achievements.

In addition to the influence of individual components of the SCT, the interplay among these components may also be related to exercise addiction. In particular, OCPD traits and self-efficacy may interact with each other to influence addictive exercise behaviors. For example, individuals with many OCPD traits may be motivated to engage in excessive levels of exercise to meet thinness or masculinity goals. If these individuals also have high levels of self-efficacy, they may be more prone to set unrealistically high goals in exercising and persist in excessive exercise despite injuries and/or interferences in functioning. Indeed, research has found that exercise-addicted individuals refused to stop their exercise regimes even when they sustained injuries or experienced disruptions to their professional and personal lives [5]. In contrast, for individuals with a combination of many OCPD traits and low self-efficacy, they are likely to discontinue their exercise when they feel incapable of maintaining their regimes. Furthermore, they may be more disturbed by and unable to cope with the possibility of failure, triggering even more negative emotions which maintain their belief that they are unable to reach their goals. The perfectionistic facet of OCPD may also contribute to an avoidance of exercise in a bid to avoid situations that force one to attempt to meet their own perfectionistic standards [55,56]. This self-perpetuating cycle of self-doubt and negativity thus results in a failure to maintain the exercise behaviors, especially since exercise is an activity requiring significant motivation and effort [35]. Therefore, the present study hypothesized that a combination of higher OCPD traits and lower self-efficacy may result in a lower risk of exercise addiction.

### 1.4. The Present Study

Early adulthood is a time where there tends to be a growing emphasis on the importance of one’s appearance. Whether due to an increasing focus on romantic relationships [57,58], pressure to adhere to sociocultural expectations of how one should look, or pressure regarding what a healthy body should look like [37,39,40], young adults are becoming increasingly conscious of their body image. This is corroborated by research showing that body dissatisfaction tends to increase from adolescence to young adulthood [59]. Considering that there is a positive association between exercise and body image [60], it is likely that young adults may engage in increasing amounts of exercise to increase body satisfaction by improving their body image. Indeed, sociocultural pressures to lose weight and build muscle were shown to lead to greater body image disturbances, which is in turn related to excessive exercise in adolescents and young adults [61].

Furthermore, some studies have found that a younger age tends to be associated with, and significantly predicts, a greater risk of exercise addiction [19,24]. The rates of exercise addiction also tend to be slightly higher in college populations [23] compared with the general adult population [22]. As such, these young adults seem to be more vulnerable to exercise addiction compared with other populations. Thus, this study aims to address the gaps in the literature to elucidate possible risk factors of exercise addiction and investigate how these factors interact within a population that is likely at a higher risk of exercise addiction.

As guided by the SCT and reviews of the available literature, it was hypothesized that:A higher level of OCPD traits will be related to a higher risk of exercise addiction.Higher levels of self-efficacy will be related to a higher risk of exercise addiction.Self-efficacy will moderate the association between OCPD traits and exercise addiction. In particular, individuals with more OCPD traits and higher self-efficacy will be more vulnerable to exercise addiction compared with individuals with more of OCPD traits but lower self-efficacy.

## 2. Materials and Methods

### 2.1. Participants

This study was approved by the Institutional Review Board of the second author of this paper. The inclusion criteria were full-time undergraduate students enrolled in universities in the United States and aged between 18 and 25 years old.

The minimum required sample size for the analyses was calculated using G*power [62,63]. For hypotheses 1 and 2, the a priori test was based on a pre-set power (1 − β = 0.95), a small effect size (f^2^ = 0.02), and α = 0.05, with four predictors (age, sex, OCPD traits, and self-efficacy). This resulted in a required sample size of 776. For hypothesis 3, the a priori test was based on a pre-set power (1 − β = 0.95), a small effect size (f^2^ = 0.02), and α = 0.05, with five predictors (age, sex, OCPD traits, self-efficacy, and interaction of OCPD traits and self-efficacy), and the minimum sample size was calculated to be 652. Hence, the total sample size of 1228 undergraduate students exceeded the minimum sample size required of 776.

Students were recruited from different university departments in different regions of the United States, consisting of 606 males and 622 females. The mean age of the students was 21.69 years old.

### 2.2. Procedure

A crowdsourcing Internet marketplace, Amazon Mechanical Turk, sent out nation-wide email invites to recruit individuals who fit the inclusion criteria. Hyperlinks to the Web survey were sent to those who consented to participate in the study. The Web survey took about 15–20 min to complete. No personal identifiable information was asked in the survey. Upon completion of the survey, participants were given a code to claim their participation fee of USD 2.5 from the Internet marketplace. The completion rate of the Web survey was 84.2%.

### 2.3. Measures

Revised Exercise Addiction Inventory (EAI-R). The EAI-R was used to assess the risk of exercise addiction [1]. The EAI-R was derived from the EAI and consists of 6 items that were designed based on the components model of behavioral addiction [1,2]. The only difference between the EAI and the EAI-R is the change from a 5-point to a 6-point scale [1]. In the EAI-R, ‘1’ signifies ‘Strongly disagree’ and ‘6’ signifies ‘Strongly agree’. Examples of items include ‘Exercise is the most important thing in my life’ and ‘Over time I have increased the amount of exercise I do in a day’.

The EAI-R was found to have very good psychometric properties above the EAI [1]. The original EAI was also shown to be a valid and reliable tool with good psychometric properties in identifying individuals at risk of, or affected by, exercise addiction in Western student populations, including the United States; also, the internal consistencies were greater than 0.72 in these studies [1,3,22]. Out of a maximum of 36 points, a score of 29 and above is the cut-off score indicating that individuals are at risk of exercise addiction. A score from 0 to 15 designates a person as asymptomatic, while a score of 16 to 28 designates a person as symptomatic. For the present study, the internal reliability was 0.82.

General Self-Efficacy Scale (GSES). The GSES was used to measure self-efficacy in the current study [64]. It consists of 10 items rated on a 4-point Likert scale, with ‘1’ signifying ‘Not at all true’ and ‘4’ signifying ‘Exactly true’. Originally created in 1979 by Jerusalem and Schwarzer, the GSES has now been adapted and translated into many different languages and has good psychometric properties when validated on various populations, including that of the current study population [64,65]. Internal consistencies have been found to range from 0.75 to 0.94 for prior studies [66] and were 0.88 for the present study.

Obsessive Compulsive Personality Disorder (OCPD) traits. OCPD traits were assessed with the 8-item OCPD subscale in the self-report Personality Diagnostic Questionnaire-4th Edition Plus (PDQ-4+) [67]. The PDQ-4+ OCPD items are consistent with the criterion of OCPD listed in Section II of the DSM-IV and DSM-5, with each question corresponding directly to a diagnostic criterion [68]. Participants answered either ‘true’ or ‘false’ to each item, and the overall score was computed by summing up all ‘true’ items. Higher scores represent a higher level of OCPD traits up to a maximum score of 8. The PDQ-4+ has been shown to have acceptable psychometric properties [68,69,70,71], even when used in non-clinical populations, including that of Western undergraduate populations [70,71]. In addition, OCPD traits as depicted in PDQ-4+ have also been shown to reflect the OCPD criteria in the proposed DSM-5 model for personality pathology reasonably well, extending the validity and utility of the PDQ-4+ [70]. For the present study, the internal reliability of this scale was 0.66.

### 2.4. Statistical Analyses

Statistical analyses were conducted with SPSS v23 software. Descriptive statistics were calculated. The associations between age, sex, exercise addiction, OCPD traits, and self-efficacy were determined using bivariate correlations. The prevalence estimates of exercise addiction were also calculated based on the EAI-R’s cutoff scores. Chi-square tests were then used to determine if there were any sex differences in the prevalence estimates. A multiple regression analysis was used to determine whether more OCPD traits and high self-efficacy were predictors of exercise addiction after controlling for age and sex. The moderation effect of self-efficacy on the relationship between OCPD traits and exercise addiction was then examined with moderation analyses using the PROCESS macro for SPSS [72].

## 3. Results

### 3.1. Preliminary Analyses

Table 1 and Table 2 show the descriptive statistics and bivariate correlations among major variables. A greater risk of exercise addiction was associated with both a male sex (*p* < 0.001) and higher levels of self-efficacy (*p* < 0.001). In addition, higher levels of self-efficacy were also associated with males (*p* < 0.001). Prevalence estimates of exercise addiction are shown in Table 3. Based on the cut-off scores of the EAI-R, 2.9% (*n* = 35) of college students were at risk of exercise addiction, while a further 59.4% (*n* = 730) reported symptoms of exercise addiction. In total, 37.7% (*n* = 463) of college students were classified as asymptomatic. Chi-square tests showed that there was an association between sex and the risk of exercise addiction (*χ*^2^ = 44.80, *p* < 0.001, Cramer’s *V* = 0.19). Post hoc tests were then conducted to test for the sex differences within the three groups (asymptomatic, symptomatic, at risk), and the significance level was adjusted based on Bonferroni correction, resulting in an adjusted *p*-value of 0.016. The post hoc tests demonstrated that males were more likely to be at risk of exercise addiction (*χ*^2^ = 4.83, *p* < 0.001) or have symptoms of exercise addiction (*χ*^2^ = 10.61, *p* = 0.001) as compared with females. However, there was no significant sex difference in the asymptomatic group (*χ*^2^ = 29.57, *p* = 0.028).

### 3.2. Results of Regression Analyses

A multiple regression analysis was conducted to test whether OCPD traits and self-efficacy predicted a greater risk of exercise addiction, as shown in Table 4. Age and sex were entered into the model in the first step, while OCPD traits and self-efficacy were entered in the second step. All predictors accounted for 8.0% of the variance in exercise addiction in model 2, *R*^2^ = 0.08, *F* (4, 1223) = 26.71, *p* < 0.001, while model 1 accounted for 4.1% of the variance in exercise addiction, *R*^2^ = 0.04, *F* (2, 1225) = 25.99, *p* < 0.001. A higher prevalence of OCPD traits (*b* = 0.44, β = 0.14, *p* < 0.001) and self-efficacy (*b* = 0.23, β = 0.17, *p* < 0.001) were predictive of a greater risk of exercise addiction.

### 3.3. Self-Efficacy as a Moderator

Moderation analyses were carried out with age and sex as covariates. Results showed that self-efficacy was a significant moderator of the relationship between OCPD traits and exercise addiction, *b* = 0.06, 95% CI [0.023, 0.088], *t* (1222) = 3.34, *p* < 0.001. As shown in Figure 1 below, for those with low levels of self-efficacy, there was no significant relationship between OCPD traits and exercise addiction, *b* = 0.19, *t* (1222) = 1.68, *p* = 0.09. However, for those with high levels of self-efficacy, a positive relationship between OCPD traits and exercise addiction was identified, *b* = 0.72, *t* (1222) = 6.02, *p* < 0.001.

In addition, sex (*b* = −2.36, 95% CI [−3.05, −1.67], *t* (1222) = −6.69, *p* < 0.001) was found to be a significant covariate in the moderation analysis. Hence, the sample was also separated by sex for further analysis. Results showed that there was no moderation effect of self-efficacy in males, *b* = 0.04, 95% CI [−0.004, 0.081], *t* (602) = 1.77, *p* = 0.077. However, a significant interaction effect between OCPD traits and self-efficacy on exercise addiction was found in females, *b* = 0.08, 95% CI [0.025, 0.128], *t* (618) = 2.92, *p* = 0.004. As shown in Figure 2, for females with low levels of self-efficacy, there was no relationship between OCPD traits and exercise addiction, *b* = −0.02, *t* (618) = −0.11, *p* = 0.91. However, for females with high levels of self-efficacy, there was a positive relationship between OCPD traits and exercise addiction, *b* = 0.70, *t* (618) = 3.84, *p* < 0.001.

## 4. Discussion

In the present study, 59.4% of the surveyed college students showed symptoms of exercise addiction, and a further 2.9% of them were found to be at risk of exercise addiction according to the EAI-R [1]. This may be due to social interactions among young adults, which may influence the way they feel about their own bodies [73]. The exposure to thin and muscular bodies on social media may also influence one’s belief of what constitutes an ideal body [37,40]. Therefore, an increase in exercise as a way of altering one’s body shape allows one to feel better about oneself [61]. As compared with females, males were more likely to either be at risk of or express symptoms of exercise addiction. This corroborates the existing literature demonstrating that being male is associated with a higher risk of exercise addiction [19,24,25]. This may potentially reflect a greater susceptibility towards sociocultural pressures in males as compared with females [74,75].

One of the aims of this study was to identify whether OCPD traits and self-efficacy are significant predictors of exercise addiction. Results showed that, as hypothesized, higher levels of OCPD traits were related to a greater risk of exercise addiction. This is in line with earlier research linking the traits of obsessive compulsiveness to a higher risk of exercise addiction [31,76]. Furthermore, it was found that higher levels of self-efficacy significantly predicted a greater risk of exercise addiction, corroborating previous studies [34]. While this is seemingly at odds with the generally adaptive role of self-efficacy, the present results highlight the fact that, unlike most of the other maladaptive behaviors, such as substance abuse, exercise addiction involves a significant amount of self-control and effort. Without being able to persevere in the face of adversity, individuals with a low level of self-efficacy are more likely to quit at the first sign of difficulty. Hence, it is likely that only individuals with high levels of self-efficacy will be able to persist in their efforts to maintain rigorous exercise regimes.

Researchers and this study have postulated that exercise addiction depends on the interaction between self-efficacy and other factors [35]. The present study adds to the literature on the role of self-efficacy as a moderator of the relationship between OCPD traits and exercise addiction. The findings showed that a higher prevalence of OCPD traits was significantly related to a greater risk of exercise addiction, but only at high levels of self-efficacy. This is in line with the SCT [26,27] that self-efficacy determines whether or not exercise addiction develops through influencing one’s thoughts, actions, and motivations. When many OCPD traits are coupled with high levels of self-efficacy, individuals tend to put in more effort in their exercise behaviors, increasing their risk of exercise addiction. However, there was no relationship observed between OCPD traits and risk of exercise addiction at low levels of self-efficacy. This may be related to the fact that exercise is an effortful activity. Individuals with a low level of self-efficacy may choose not to engage in exercise at all, regardless of the prevalence of OCPD traits.

Furthermore, the present study also demonstrated that the moderating effect of self-efficacy between OCPD traits and exercise addiction is evident only in females. This may be due to differences in perceived social pressure and ideal body beliefs. Males are expected to engage in exercise both culturally and socially, as they are more focused on increasing muscle mass as opposed to maintaining a slim body shape, which can be achieved through dieting, for females [73,74]. In addition, pressure from dating partners and peers was also shown to have a greater influence on activity levels and exercise commitment in males as opposed to females [74,75]. Moreover, while dieting can be used as an alternative to achieve one’s desired appearance, males are less willing to undertake dieting as it is commonly seen as an activity for females [76,77,78]. Thus, exercise may be the most effective and potentially the sole means by which males can attain their objectives. Hence, males are more likely to engage in addictive exercise behaviors regardless of their self-efficacy.

While exercise addiction is not a widespread occurrence, early identification and intervention is necessary to reduce negative physical and mental health consequences. The present study showed that the SCT [26,27] is applicable in understanding the development and maintenance of exercise addiction. Unlike other addiction interventions which attempt to increase self-efficacy as a way to change maladaptive behaviors [79], increasing self-efficacy should be used with caution when crafting interventions to reduce exercise addiction, especially among females. Instead, psychoeducation can be used to educate individuals as to how their strong belief in their own capabilities can result in maladaptive exercise behaviors, especially if they have certain personality traits such as perfectionism or OCPD traits.

The present study is one of the first to investigate the factors of OCPD traits and self-efficacy in relation to exercise addiction. Hence, the results obtained should be replicated in other studies. Self-efficacy should also be examined in more diverse populations to investigate if it plays the same role as it does here in young adults. There were certain limitations regarding the measurement scales used in the study. First, the measurement of OCPD traits is relatively broad-based, such that specific facets like perfectionism or rigidity were not assessed in detail. Similarly, only general self-efficacy was measured, rather than different aspects of self-efficacy such as task or scheduling self-efficacy. There was also no external verification of self-reported exercise behaviors with the online survey. In addition, data were not collected on co-occurring health and mental conditions, such as eating behaviors. The presence of eating disorders or food addiction, for instance, may affect the relationship between OCPD traits, self-efficacy, and exercise addiction. The results of this study were obtained via self-report measures that are cross-sectional in nature; hence, caution should be taken in interpreting the results. In particular, causal relationships cannot be inferred. The present study surveyed college students only, and its findings may not be generalizable to other populations, such as teenagers and working adults.

## 5. Conclusions

This study estimated the prevalence of exercise addiction in US college students and revealed the moderating role of self-efficacy in the relationship between OCPD traits and exercise addiction risk. The prevalence estimate of US college students at risk of exercise addiction is 2.9% (males: 4.0%; females: 1.8%). Also, males are more prone to experience symptoms of or be at risk of exercise addition. Furthermore, the results showed that OCPD traits are associated with exercise addiction only at high levels of self-efficacy in females, but not in males. These findings thus have implications that public education and interventions for exercise addiction should attend to regarding the interplay between personality factors and sex.

## Figures and Tables

**Figure 1 behavsci-13-00857-f001:**
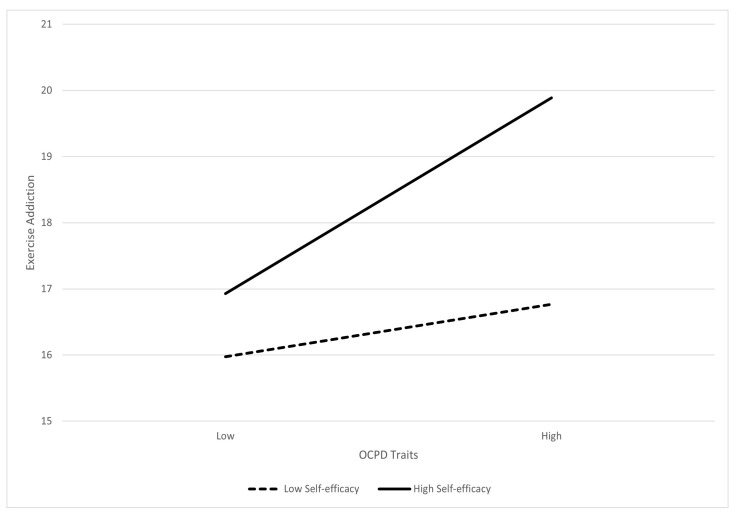
Moderation by self-efficacy on OCPD traits and exercise addiction.

**Figure 2 behavsci-13-00857-f002:**
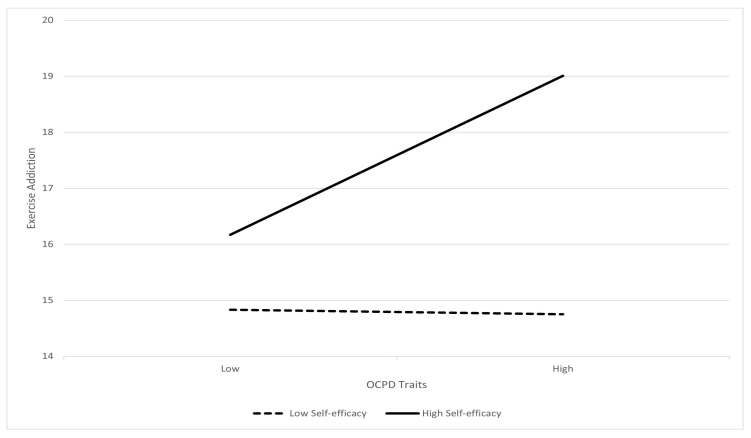
Moderation by self-efficacy on OCPD traits and exercise addiction—females.

**Table 1 behavsci-13-00857-t001:** Descriptive statistics of major variables.

	Internal Consistency	Total *M* (*SD*)	Male *M* (*SD*)	Female *M* (*SD*)	Sex Difference
Age	-	21.68 (2.00)	21.58 (2.05)	21.78 (1.95)	*t* (1226) = −1.74 *, Cohen’s *d* = 0.10
EA	0.82	17.30 (6.41)	18.55 (6.12)	16.07 (6.45)	*t* (1225.11) = 6.92 ***, Cohen’s *d* = 0.39
OCPD	0.66	2.86 (2.06)	2.83 (2.09)	2.90 (2.04)	*t* (1226) = −0.584, Cohen’s *d* = 0.03
SE	0.88	30.86 (4.72)	31.34 (4.72)	30.40 (4.67)	*t* (1226) = 3.52 **, Cohen’s *d* = 0.20

Note: EA = exercise addiction, OCPD = obsessive compulsive personality disorder traits, SE = self-efficacy. * *p* < 0.05. ** *p* < 0.01. *** *p* < 0.001.

**Table 2 behavsci-13-00857-t002:** Bivariate correlations among major variables.

	1	2	3	4	5
Age	-				
Sex	0.05	-			
EA	0.05	−0.19 ***	-		
OCPD	0.00	0.02	0.11 ***	-	
SE	0.03	−0.10 ***	0.16 ***	−0.17 ***	-

Note: EA = exercise addiction, OCPD = obsessive compulsive personality disorder traits, SE = Self-Efficacy. *** *p* < 0.001.

**Table 3 behavsci-13-00857-t003:** Prevalence of risk of exercise addiction.

	At Risk *n* (% Subgroup)	Symptomatic *n* (% Subgroup)	Asymptomatic *n* (% Subgroup)
Total (*n* = 1228)	35 (2.9%)	730 (59.4%)	463 (37.7%)
Male (*n* = 606)	24 (4.0%)	409 (67.5%)	173 (28.5%)
Female (*n* = 622)	11 (1.8%)	321 (51.6%)	290 (46.6%)
Sex Differences	*χ*^2^ = 4.83 **	*χ*^2^ = 10.61 *	*χ*^2^ = 29.57

Note: * *p* < 0.016, ** *p* < 0.001.

**Table 4 behavsci-13-00857-t004:** Multiple regression analysis of OCPD traits and self-efficacy.

Model		Unstandardized Coefficients		Standardized Coefficients	*t*	*p*	*F*	*df*	*R* ^2^	*R*^2^ Change	*p*
		*B*	SE	β							
1							25.99	2, 1225	0.041	0.041	0.000 ***
	Age	0.18	0.09	0.06	2.01	0.045 *					
	Sex	−2.52	0.36	−0.20	−0.20	0.000 ***					
2							26.71	4, 1223	0.080	0.040	0.000 ***
	Age	0.16	0.09	0.05	1.80	0.072					
	Sex	−2.33	0.35	−0.18	−6.59	0.000 ***					
	OCPD	0.44	0.09	0.14	5.11	0.000 ***					
	SE	0.23	0.04	0.17	5.96	0.000 ***					

Note: OCPD = obsessive compulsive personality disorder traits, SE = self-efficacy. * *p* < 0.05, *** *p* < 0.001.

## Data Availability

All data relevant to the present study are available from the corresponding author upon reasonable request.

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
