# Peer review of "The Associations between Obsessive Compulsive Personality Traits, Self-Efficacy, and Exercise Addiction"

_behavsci, 2023, doi:10.3390/bs13100857_

Round 1

Reviewer 1 Report

see file attached

Basically quite good, but some points it seems a bit awkward. Please have a native speaker check it again

Author Response

Reviewer Comment                                                                                             

  1. in l. 44, begin to present the prevalence rates of sports addiction . Normally they are less than 10%, but in the Australian sample 41% were found. How can that be? Can you explain this large difference?

****The difference in prevalence rate was likely due to the use of a different measure of exercise addiction. We have amended the article to explain this difference for clarity (bottom of page.1 to top of page 2)

  1. In line 157, the hypotheses are presented. Why did you not include the "age" factor? In the theoretical discussions, you often point out that age has an influence. In the evaluation you take this into account, why not in the hypotheses?

**** We wanted to focus solely on college students as a population more vulnerable to exercise addiction when examining the factors of OCPD traits and self-efficacy. As a college population, their age tends to be similar as well. Thus, we did not expect age to be a significant factor in our sample, and did not include it in our hypotheses. This is corroborated by the bivariate analyses where age did not correlate significantly with any other examined factor.

  1. Can you please give some examples of the items of the EAI-R? This would help readers understand what exactly is being measured

**** Some examples of the items of the EAI-R have been included for clarity (line 202-203)

  1. In the manuscript, they often switch between the term sex and gender, but they always mean the same thing, right? Please stay consistent with one term

**** We have edited the text to only use the term "sex" according to the journal's guidelines on sex and gender.

  1. Table 2: Did you also test the difference in the symptomatic and asymptomatic expression groups for significance? This could provide further valuable information for the discussion (and should then also be mentioned there).

*****We have made amendments to the article to better clarify the results of the chi-square test, as well as the post-hoc tests used to determine the exact differences in sex among the at risk, symptomatic, and asymptomatic groups. (top of p.6, line 255-261). This finding was also included in the discussion in relevant places.

  1. Perhaps you could also present the results of the regression in a table, that would be clearer. And please be sure to show the difference in R-squared between the different models

****Table 4 is added to present the results of the regression as suggested. R-squared change between the 2 models have also been added to the table (p.4).

This article is filled with many typos. Therefore, it is necessary to fully adapt the article to the English language.

****The manuscript has been checked and edited for English Language use.

We have reviewed the article again in detail, and have rectified the language errors that were previously present to conform with the English language rules. The article was also edited to allow for better flow where possible.

Reviewer 2 Report

Title – The Associations among Obsessive-Compulsive Personality Traits, Self-Efficacy, and Exercise Addiction

This study aims to address the literature gaps, elucidate possible risk factors of exercise addiction, and investigate how these factors interact within a population that is likely at a higher risk of exercise addiction.

I have thoroughly reviewed the manuscript and would like to make some suggestions for your review.

This article is filled with many typos. Therefore, it is necessary to fully adapt the article to the English language rules.

Other points

Line 165: Sample, the sample size calculations should be presented, please use the G*power to justify this issue.

Line 224 -226: Write the names of the variables in their abbreviated form in Table 1 - 2, and write the extended versions of the abbreviations at the end of the table.

Line 224: table 1 add effect size values to indicate the difference between variables

Line 322: It is not usual to cite any literature in the conclusions. This section should summarize your findings. Please delete it.

This article is filled with many typos. Therefore, it is necessary to fully adapt the article to the English language rules.

Author Response

Line 165: Sample, the sample size calculations should be 2presented, please use the G*power to justify this issue.

****The methodology has been amended to include the sample size calculations using G*power as justification for the current sample (p.4, line 176-184)

Line 224 -226: Write the names of the variables in their abbreviated form in Table 1 - 2, and write the extended 2versions of the abbreviations at the end of the table.

******Apart from the variables of age and sex, we have edited the variable names to only include their abbreviated form in Tables 1-2, and the extended form at the end of the table (p. 6)

Line 224: table 1 add effect size values to indicate the difference between variables

****We have included the effect size values for sex differences between each variable in Table 1 as suggested.

Line 322: It is not usual to cite any literature in the conclusions. This section should summarize your findings. Please delete it.

**** We have updated the conclusion to summarize our findings, and deleted the citations as suggested. (p.10).

This article is filled with many typos. Therefore, it is necessary to fully adapt the article to the English language rules.

****We have reviewed the article again in detail  and have rectified the language errors that were previously present to conform with the English language rules. The article was also edited to allow for better flow where possible.

Reviewer 3 Report

We are faced with an interesting article that tries to establish the relationships between OCPD, self-efficacy and exercise addiction. At first I have to comment that I think it is a very original and interesting paper.

I would like to make some considerations so that the authors can improve the paper.

Introduction:

I find the introduction very clear, well structured, presenting the most relevant aspects and a very pleasant read. In my opinion, perhaps some more current references on the different aspects were missing.

Measures: 

The authors present the measurement instruments citing "For the present study." Does this mean that internal validity was calculated specifically for this study? Was the same sample used for this or was a different sample used? Had the measurement instruments been previously validated for the populations participating in this study? Could CFA or any other more relevant data on said instruments be presented?

Results:

In line 222 the authors show that they have obtained significant differences between males and females, using the Chi-square test. In my opinion, this test should be included in the Statistical Analyzes section.

The results would gain a lot of weight in general if, beyond the level of significance, the authors also provide the power of the effects obtained (easily calculable, for example, through the G*power software).

Conclusion:

In my opinion, the conclusions should focus on answering the AIM and Hypotheses of the study. The authors continue discussing in the conclusions. They should include this information in the discussion and make a specific clear and concrete section on the conclusions.

Format Error:

Line 37. Double space after the point.

Line 119. Double space after the point.

Author Response

I find the introduction very clear, well structured, presenting the most relevant aspects and a very pleasant read. In my opinion, perhaps some more current references on the different aspects were missing.

****Thank you. We have reviewed the currently available literature and have included more recent references where possible.

For the present study." Does this mean that internal validity was calculated specifically for this study? Was the same sample used for this or was a different sample used?

****Yes, internal reliability was calculated specifically for this study using the same sample. The EAI-R (or its predecessor, EAI) and the GSES have previously been validated for the US college student population. Studies have also shown that the PDQ-4+ has adequate psychometric properties when used with undergraduate populations in Western countries, and it reflects the OCPD criteria even in the new proposed DSM-5 model of personality pathology. The measures section has been edited to reflect this information more clearly to show the validity of the measures used. (p.5)

In line 222 the authors show that they have obtained significant differences between males and females, using the Chi-square test. In my opinion, this test should be included in the Statistical Analyzes section.

****We moved the sex differences in the statistical analyses section (line 28-239).

The results would gain a lot of weight in general if, beyond the level of significance, the authors also provide the power of the effects obtained (easily calculable, for example, through the G*power software).

****While we understand that information on the power of the effects observed may help to back up our findings, we also note that much of the current literature cautions the use of post-hoc power analyses as power is determined by the p value used. Hence, we have updated the article to include information on how we used the G*power software to conduct an a priori test to generate the minimum sample size required to observe a small effect size (if present), as further justification for our results instead.

In my opinion, the conclusions should focus on answering the AIM and Hypotheses of the study. The authors continue discussing in the conclusions. They should include this information in the discussion and make a specific clear and concrete section on the conclusions.

**** We have re-arranged the discussion previously present within the conclusion to the Discussion section. The conclusion has also been amended to clearly answer the aims and hypotheses of the study instead.

Round 2

Reviewer 2 Report

Corrections in the text have been made appropriately.

Reviewer 3 Report

The suggestion was made. For me this paper is ok.